# Timing of Deep and REM Sleep Based on Fitbit Sleep Staging in Young Healthy Adults under Real-Life Conditions

**DOI:** 10.3390/brainsci14030260

**Published:** 2024-03-06

**Authors:** Charlotte von Gall, Leon Holub, Amira A. H. Ali, Simon Eickhoff

**Affiliations:** 1Institute of Anatomy II, Medical Faculty, Heinrich Heine University, 40225 Düsseldorf, Germany; mail@leonholub.com (L.H.); amira.ali@med.uni-duesseldorf.de (A.A.H.A.); 2Institute of Systems Neuroscience, Medical Faculty, Heinrich Heine University Düsseldorf, 40225 Düsseldorf, Germany; simon.eickhoff@uni-duesseldorf.de; 3Institute of Neuroscience and Medicine, Brain and Behaviour (INM-7), Research Centre Jülich, 52425 Jülich, Germany

**Keywords:** chronotype, sleep quality, REM sleep fragmentation, sleep hygiene, wearables, sleep architecture, sleep composition

## Abstract

Sleep timing is controlled by intrinsic homeostatic and circadian components. The circadian component controls the chronotype, which is defined by the propensity to sleep at a particular clock time. However, sleep timing can be significantly affected by external factors such as the morning alarm clock. In this study, we analysed the timing of deep and REM sleep as well as the composition of REM sleep using Fitbit sleep staging in young healthy adults (*n* = 59) under real-life conditions. Sleep stage percentiles were correlated with the timing of total sleep in *time after sleep onset* for the homeostatic component and in *clock time* for the circadian component. Regarding the circadian component, the phase of total sleep is most strongly associated with the phases of early deep sleep and REM sleep. Furthermore, a stronger phase relationship between deep and REM sleep with total sleep is associated with greater consolidation of REM sleep. Chronotype-dependent sleep loss correlates negatively with the strength of the phase relationship between deep sleep and total sleep. In conclusion, the interaction of the circadian component of sleep timing with the timing of sleep stages is associated with REM sleep quality. In particular, the interaction of the circadian component of sleep timing with deep sleep seems to be more vulnerable to external factors.

## 1. Introduction

Sleep plays a crucial role for performance, well-being and health in general [1,2]. In healthy adults, sleep is composed of 4–6 sleep cycles per night, each lasting around 70–110 min [3,4]. Each sleep cycle consists of a sequence of different sleep stages characterized by variations in muscle tone, brain activity patterns, heart rate and eye movements [5]. The sleep cycle progresses from light sleep to deep sleep to rapid eye movement (REM) sleep and ends with a short bout of waking. Deep sleep, also known as slow-wave sleep or N3, is most restorative for the brain and the body, including the immune system [6], while REM sleep is associated with dreaming and appears to be the sleep stage most relevant to sleep quality, cognitive performance and mood [7,8]. Deep sleep and REM sleep each make up about a quarter of the total sleep [5]. The duration of deep sleep in the sleep cycles decreases during the night, while the duration of REM sleep increases [3,4]. The sleep-wake cycle is controlled by complex neuronal networks [9], driven by internal oscillators and modulated by external oscillators [10]. These oscillators control not only sleepiness at night and alertness during the day, but also the timing, duration and composition of sleep and therefore sleep quality [10,11]. The internal oscillators control the homeostatic and the circadian component of sleep timing so that both are able to work independently of each other [11]. The main driver of the homeostatic component is sleep pressure, which builds up with increasing time awake and decreases during sleep. The circadian component, controlled by the circadian clock, defines the propensity to sleep at a particular clock time, thus the chronotype [12]. Thus, individuals differ not only in their preferred sleep *duration*, but also in their preferred sleep *time*, dependent on their chronotype [13]. 

The midpoint of sleep relative to the time elapsed since falling asleep reflects the homeostatic component of sleep timing. The midpoint of sleep relative to ambient time reflects the circadian component of sleep timing. Importantly, workday constraints affect sleep timing. The individual chronotype is therefore determined on work-free days [13]. On workdays, the morning alarm clock largely determines both sleep duration and midpoint of sleep. Thus, in particular in late chronotypes, early schedules result in discrepancies of sleep duration and midpoint of sleep between workdays and work-free days, which are defined as sleep loss and social jetlag, respectively [14]. This misalignment of biological (work-free days) and social (workdays) timing of sleep has various health consequences including impaired sleep quality and depressive symptoms [15,16,17]. 

The aim of our study was to investigate the homeostatic and circadian components of the timing of deep and REM sleep in humans under real-life conditions based on Fitbit sleep staging. Sleep staging by algorithms based on heart rate and movement patterns using wrist-worn wearables is less accurate than the gold standard, polysomnography [18,19,20], but has high potential and the advantage for long-term studies under real-life conditions. Results from longitudinal studies on sleep structure generated by Fitbit proprietary algorithms [21], combined with questionnaires of young healthy adults by Weiss et al. [22] and our group [23,24], support the hypothesis of an interaction of subjective sleep quality, chronotype, objective sleep architecture, psychosocial stress and depressive symptoms. In this study, we used the same data set as our previous studies [23,24] to calculate the intraindividual (within-subject) correlation coefficients of the deep and REM sleep percentiles with the midpoint of sleep in relation to the time elapsed since falling asleep and in relation to ambient time. This allows conclusions about the strength and direction of the relationships between the homeostatic and circadian components of sleep timing and the timing of deep and REM sleep. In addition, the intraindividual correlation coefficients of the 50 deep and REM sleep percentiles with the midpoint of sleep were correlated with the score for subjective tiredness as well as with REM sleep proportion and fragmentation. This allows conclusions about the strength and direction of relationships between sleep stage timing and sleep quality. The data were analyzed separately for workdays and work-free days. This allows conclusions about the effect of workday constraints. 

## 2. Materials and Methods

### 2.1. Ethics, Study Design and Sample 

The background material in this section is a nearly verbatim adaptation of Section 2.1 of [23]. This study was performed in agreement with the Declaration of Helsinki ethical requirements, and approved by the Research Ethics Committee of the Heinrich Heine University Medical Faculty (ChronoSleep study consent number: 2019-3786). Informed consent was obtained after the nature and possible consequences of the study were explained. Exclusion criteria were (1) age below 18 years, (2) shift work, (3) work on weekends, (4) chronic diseases including sleep disorders and (5) chronic medication including sleep medication. Part of the data used for this study came from the same data set of medical students in their first year as described earlier [23,24]. Briefly, volunteers received a pseudonym and were equipped with a Fitbit Inspire multisensory (motion and heartrate) sleep tracking device and asked to wear it for 90 days, as continuously as possible, especially at nights. The proprietary Fitbit algorithms detect sleep stage based on heartrate and activity patterns [21]. The data collection period includes 64 weekdays, defined as workdays, and 26 weekend days, defined as work-free days. After the data collection period, the volunteers were asked to actively participate in the study by completing the online questionnaire and then authorizing the transfer of sleep data from Fitbit to our study server. Of the 90 participants who met the criteria, 31 were excluded because they did not authorize the data transfer or because of missing sleep data for more than 35 days. This resulted in a final sample size of *n* = 59 (73% females). In this sample, an average of 82.4 (±9.7) days of sleep data were recorded. A detailed description of the sample characteristics has been published previously elsewhere [23,24]. The average age was 21.18 (±2.36) years. The data collection period between 25 June 2021 and 19 May 2022 fell within the COVID-19 pandemic. Most of the lectures, which were made available as screencasts, were watched by the students according to their own schedule, but usually in the morning, as on-site courses usually began at 11 a.m. Based on the questionnaire, we know that all subjects used an alarm clock on workdays, while only 15 subjects used one on work-free days. In addition, confounding factors for sleep timing on work-free days, such as social gatherings, were largely reduced due to the general closure of most of the restaurants, bars and nightclubs.

### 2.2. Data Analyses

For graphical representation of longitudinal Fitbit sleep stage data, double-plotted actograms were created using the ClockLab toolbox (version 6.1.10, Actimetrics, Wilmette, IL, USA) at a 60 s resolution. 

A customized Python (version 3.10) application [23] was used to calculate the following variables based on the individual longitudinal Fitbit sleep stage recordings:
The midpoints of sleep in *minutes after sleep onset* and *clock time*, reflecting the phase of sleep regarding the homeostatic and the circadian component, respectively.The sleep stage percentiles, indicating at what times 25%, 50%, 75% and 100% of the respective sleep stage was reached. The sleep stage percentiles were also determined in relation to *minutes after sleep onset* and *clock time*, reflecting the phases regarding the homeostatic and circadian component, respectively.To investigate the strength of intraindividual phase relationships between total sleep and the sleep stages, bivariate correlations between the midpoint of sleep and the sleep stage percentiles were computed for Spearman’s rank coefficients with a 95% confidence interval. The resulting intraindividual correlation coefficients (ICC) were used as a measure of the strengths of the phase relationship. The larger the ICC, the stronger the phase relationship.Sleep duration.Proportion of REM sleep in percentage of total sleep.Fragmentation index of REM sleep (RFI) as the number of transitions from REM sleep to any other sleep stage per hour of REM sleep.

Variables I.–IV. were analyzed separately for workdays and work-free days, to account for workday constraints such as the start of work in the morning, which often requires waking up with an alarm clock. Sleep loss was calculated based on the difference in sleep duration between workdays and work-free days.

Normality of the variables was tested by the D’Agostino–Pearson test. As the variables were partly not normally distributed, a Wilcoxon test was applied to detect differences between variables using Graph Pad Prism software (Version 7.01). *p*-values < 0.05 were considered statistically significant. To investigate the direction of interindividual (between-subject) relationships between variables, bivariate correlations were computed for Spearman’s rank coefficients with a 95% confidence interval using Graph Pad Prism software (Version 7.01). The correlation coefficient was considered significantly different from zero when *p* < 0.05.

## 3. Results

### 3.1. Within-Subject Sleep Data Analyses

This subsection introduces the main intraindividual variables, for illustrative purposes, only for one representative subject. First, the percentiles of the sleep stages are related to the time of day and to the time that has elapsed since sleep onset in order to separate the circadian and homeostatic effects. Afterwards, the intraindividual correlation coefficients (ICC) of the sleep phase percentiles with the midpoint of sleep are determined as a measure of the strength of the phase relationships.

#### 3.1.1. Sleep Stage Percentiles

Actograms of longitudinal recordings of Fitbit sleep staging show that deep sleep (Appendix A) is more prevalent during early sleep, while REM sleep (Appendix A) is more prevalent during late sleep. The sleep stage percentiles were determined in relation to *minutes after sleep onset* (Figure 1a–c) and *clock time* (Figure 1d–f), reflecting the homeostatic and circadian effect, respectively. 

#### 3.1.2. Intraindividual Correlation Coefficients (ICC) of the Sleep Phase Percentiles with the Midpoint of Sleep as a Measure of the Strength of the Phase Relationships

The ICC of the stage percentiles and midpoint of sleep in minutes after sleep onset (Figure 2a) were positive and highest for the later REM sleep percentiles. This indicates the strongest phase relationship between total sleep and late REM sleep in relation to the homeostatic component, in this subject. 

The ICC of the stage percentiles and midpoint of sleep in clock time (Figure 2b) were highest for the earlier deep sleep percentiles. This indicates the strongest phase relationship between total sleep and early deep sleep in relation to the circadian component, in this subject. 

Moreover, most of the ICC were lower on workdays than on work-free days (Figure 2). This suggests that work-day constraints affect the phase relationship between total sleep and the sleep stages. 

In addition, using the ICC, the phase relationship between deep and REM sleep were analyzed (Appendix A). These ICC were lower than the ICC of the sleep stage percentiles with the midpoint of sleep (Figure 2). This indicates a weaker phase relationship between deep and REM sleep than between total sleep and the sleep stages in this subject.

### 3.2. Between-Subject Sleep Data Analyses 

In this subsection, the distribution of the ICC as a measure of the phase relationship between total sleep and the sleep stages is shown for all subjects. In addition, the relationships between the ICC and measures of sleep quality are shown. 

#### 3.2.1. Distribution of the ICC 

The distribution of the ICC of the midpoint of sleep with the sleep stage percentiles in *minutes after sleep onset* of all subjects is shown in Figure 3. The ICC of the midpoint of sleep with the later deep sleep (Figure 3a) and REM sleep (Figure 3b) percentiles are higher than the ICC of the midpoint of sleep with the earlier sleep stage percentiles. Thus, the phase relationship between the sleep stages and total sleep increases with the time elapsed since sleep onset. Moreover, the ICC of the midpoint of sleep and the mid-to-late deep sleep percentiles are higher on work-free days than on workdays (Figure 3a). This suggests that workday constraints interfere with the temporal relationship between deep sleep and total sleep with regard to the homeostatic component. 

The respective ICC of the deep and REM sleep percentiles show a similar pattern (Appendix A) but are lower than the ICC of the sleep stage percentiles with the midpoint of sleep. This indicates a stronger temporal relationship between the sleep stages and total sleep than between deep and REM sleep. 

The distribution of the ICC of the midpoint of sleep and the sleep stage percentiles in *clock time* of all subjects is shown in Figure 4. The ICC are highest for the midpoint of sleep and the earlier deep sleep percentiles (Figure 4a). This indicates the strongest phase relationship between earlier deep and total sleep with regard to the *circadian* component. Moreover, the ICC of the midpoint of sleep and the early-to-mid deep sleep percentiles are higher on work-free days than on workdays (Figure 4a). This suggests that workday constraints interfere with the temporal relationship between deep sleep and total sleep with regard to the circadian component. 

The ICC are equally high for the midpoint of sleep and all REM sleep stage percentiles (Figure 4b). This suggests a strong temporal relationship between REM sleep and total sleep with regard to the circadian component. Moreover, the ICC are equally high for workdays and free days (Figure 4b). Thus, the temporal relationship between REM sleep and total sleep seems to be less vulnerable to workday constraints than the temporal relationship between deep sleep and total sleep. 

The respective ICC of the deep with the REM sleep percentiles show a similar pattern (Appendix A) but are lower than the ICC of the sleep stage percentiles with the midpoint of sleep. This indicates a weaker temporal relationship between deep and REM sleep than between the sleep stages and total sleep.

#### 3.2.2. Relationship between the ICC and Measures of Sleep Quality

As a measure of the intraindividual phase relationship between the sleep stages and total sleep, the ICC of the 50 percentile of the sleep stages and the midpoint of sleep (ICC50) were used. The ICC of the 50 percentiles of REM/deep sleep and the midpoint of sleep in minutes after sleep onset, thus for the *homeostatic* component, are defined as ICC50-H. The ICC of the 50 percentiles of REM/deep sleep and the midpoint of sleep in *clock time*, thus for the circadian component, are defined as ICC50-C. The ICC50 were correlated with chronotype-dependent sleep loss, as well as REM sleep proportion and fragmentation, as promising digital markers for sleep quality [23], and tiredness upon waking on work-free days as a measure of subjective sleep quality in the absence of workday constraints [23]. 

The correlations of the ICC 50-H of deep and REM sleep with measures of sleep quality are shown in Table 1. On workdays, the ICC 50-H of REM sleep correlate positively with the proportion of REM sleep (Table 1). Thus, under workday constraints, a higher temporal relationship between REM sleep and total sleep, with regard to the homeostatic components, is associated with a higher proportion of REM sleep. 

The correlations of the ICC50-C of deep and REM sleep with measures of sleep quality are shown in Table 2. The ICC50-C of REM sleep correlate positively with the proportion and negatively with the fragmentation of REM sleep. Thus, a higher strength in the temporal relationship between REM sleep and total sleep, with regard to the circadian component, is associated with a higher consolidation of REM sleep. 

On work-free days, the ICC50-C of deep sleep correlate negatively with sleep loss and tiredness upon waking, and REM sleep fragmentation. Thus, in the absence of workday constraints, a higher strength in the temporal relationship between deep sleep and total sleep is associated with a lower discrepancy in sleep duration between workdays and work-free days, a higher subjective sleep quality and a higher consolidation of REM sleep.

## 4. Discussion

In this study, we investigated the temporal relationship of deep and REM sleep with total sleep, based on Fitbit sleep analysis, in humans under real-life conditions. 

We used the intraindividual correlation coefficients (ICC) of the deep sleep and REM sleep percentiles with the midpoint of sleep relative to the *time elapsed since falling asleep* and to *clock time*, to elucidate the strength of the phase relationships in relation to the homeostatic and the circadian component of sleep timing, respectively. Interestingly, our data suggest a stronger temporal relationship between the sleep stages and total sleep than between deep and REM sleep. In other words, the timing of REM sleep seems to be more strongly associated with the timing of total sleep than with the timing of deep sleep. 

The ICC of the sleep stage percentiles with the midpoint of sleep increase progressively with the *time elapsed since falling asleep* (Figure 5a). This suggests an increasingly strong temporal relationship between the sleep stages and total sleep with regard to the homeostatic component. This is consistent with a homeostatic forward regulatory “hourglass” mechanism that controls the timing of REM sleep episodes in rodents [25,26,27]. In contrast, with respect to *clock time*, early deep sleep is more strongly associated with total sleep than late deep sleep (Figure 5b). This suggests a stronger temporal relationship between early deep sleep and total sleep with regard to the circadian component. For all REM sleep percentiles, the correlation with total sleep relative to clock time is high. This suggests a strong temporal relationship between REM sleep and bedtime. This is consistent with the concept of tight control of REM sleep timing by the circadian process of sleep regulation, which is mainly based on experiments under artificial conditions [10,11,12,28,29,30]. However, under artificial conditions, the distortion by stress hormones, which strongly influence the circadian system, cannot be ruled out [31]. Our real-life study therefore makes an important contribution to understanding the temporal control of sleep stages.

The strength of the temporal relationship between deep sleep and total sleep seems to be higher on work-free days than on workdays. This suggests a negative impact of workday constraints on the phase relationship between deep sleep and total sleep. In other words, on workdays the alarm clock in the morning not only shortens sleep duration and advances the midpoint of sleep [23] but also interferes with deep sleep timing (this study). In contrast, the strength of the temporal relationship between REM sleep and total sleep is not significantly different between workdays and work-free days. Thus, in contrast to an abrupt phase shift of the light/dark cycle that interferes with the REM sleep architecture in rodents [32], workday constraints seem to have no significant impact on REM sleep timing in our sample. 

To examine the relevance of the temporal relationship between sleep stages and total sleep timing, we analyzed the association with measures of sleep quality. As we have shown recently, subjective tiredness on work-free days correlates negatively with the percentage and positively with the fragmentation of REM sleep based on Fitbit sleep staging [23]. This suggests a relationship between REM sleep consolidation and subjective sleep quality consistent with the role of REM sleep composition for sleep quality [7,33]. Importantly, timing, duration and quality of sleep contribute essentially to performance and emotional health [24,34]. Here we show that the ICC50-C *REM* correlate positively with REM sleep proportion and negatively with REM sleep fragmentation (Figure 5c). Thus, a higher temporal relationship of REM sleep with total sleep, relative to the circadian component, is associated with higher REM sleep consolidation. The ICC50-C *deep* on work-free days correlate negatively with REM sleep fragmentation and tiredness on waking (Figure 5c). Thus, in the absence of workday restraints, a stronger phase relationship of deep sleep with total sleep is associated with higher REM sleep consolidation and better subjective sleep quality. Moreover, chronotype-related sleep loss, and thus, a higher discrepancy between sleep duration during workdays and work-free days, is associated with a weaker temporal relationship between deep sleep and total sleep on work-free days. Thus, sleep loss is both directly [23] and indirectly (this study) associated with a reduced sleep quality. This is consistent with the current consensus statement of the National Sleep Foundation that catch-up sleep on non-working days may be beneficial [35].

Interestingly, based on animal research, higher fragmentation and lower density of REM sleep have also been associated with poor sleep quality in post-traumatic stress disorder [36,37], due to changes in the neuronal network that regulates REM sleep, including the REM-on/off cells of the pontine brainstem [38]. Our analysis of the strength of the intrinsic phase relationships between sleep and sleep stages in real-world conditions could significantly contribute to a better understanding of the interactions between sleep, circadian rhythms and mental health [39], especially as both REM [22,40] and deep sleep [40] are particularly important for mental health. Measuring the intrinsic phase relationships between sleep stages and sleep could also be used in a broader clinical context in the future. We used the Fitbit wearable because it is affordable, well validated [19] and widely used. However, other wearables could also be used for similar analysis. 

In conclusion, our study suggests that a stronger temporal relationship between deep and REM sleep with total sleep in clock time, is associated with higher REM sleep consolidation that could contribute to sleep quality. It also suggests that workday constraints and chronotype-dependent sleep loss interfere with the phase relationship between deep sleep and total sleep, which could also contribute to sleep quality. Therefore, taking the individual sleep propensity time into account in personalized sleep hygiene could help improve sleep quality.

## 5. Limitations

Sleep staging by algorithms based on heart rate and movement patterns using wrist-worn wearables is less accurate than the gold standard, polysomnography [18,19,20]. Moreover, the generalization of our study is limited because the sample is rather small, not representative of the population and has biases in many ways, such as age, education, income, biological sex, motivation to participate and the unique circumstances of the study period, which was during the COVID-19 pandemic. We cannot rule out that elevated stress during these unique circumstances might be a confounder for the circadian system. Therefore, larger cohort studies in healthy subjects and patients under non-COVID-19 conditions are needed for further validation of our results. 

## Figures and Tables

**Figure 1 brainsci-14-00260-f001:**
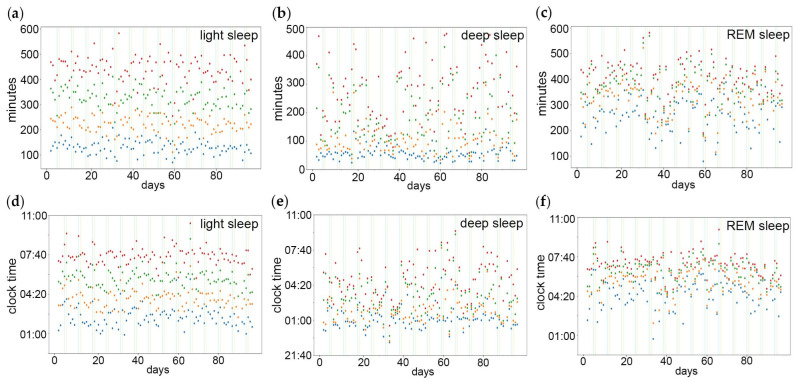
Percentiles of sleep stages in relation to the homeostatic and the circadian component of sleep timing. The percentiles indicate at what points in time 25% (blue dots), 50% (orange dots), 75% (green dots) and 100% (red dots) of (**a**,**d**) light sleep, (**b**,**e**) deep sleep and (**c**,**f**) REM sleep have been reached. The sleep stage percentiles were determined in relation to (**a**–**c**) minutes after sleep onset and (**d**–**f**) clock time, reflecting the homeostatic and circadian component, respectively. Vertical lines indicate weekend/work-free days.

**Figure 2 brainsci-14-00260-f002:**
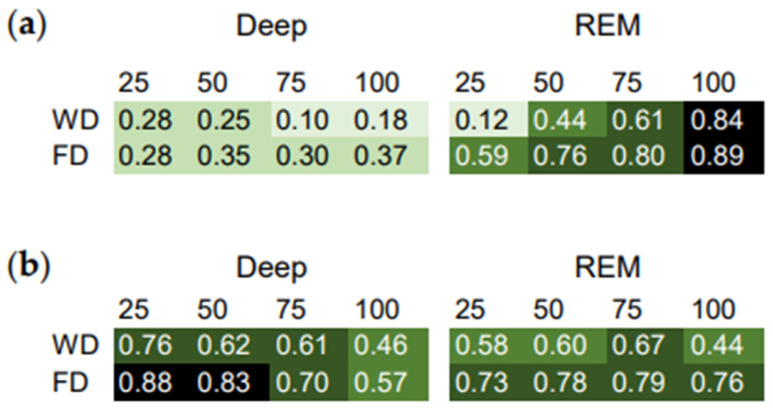
Intraindividual correlation coefficients (ICC) as a measure of the phase relationship between total sleep and the sleep stages. Correlation matrix of the sleep stage percentiles and the midpoint of sleep relative to (**a**) *minutes after sleep onset* and (**b**) *clock time*. Workdays (WD) and work-free days (FD) were evaluated separately. Higher ICC indicate stronger relationships.

**Figure 3 brainsci-14-00260-f003:**
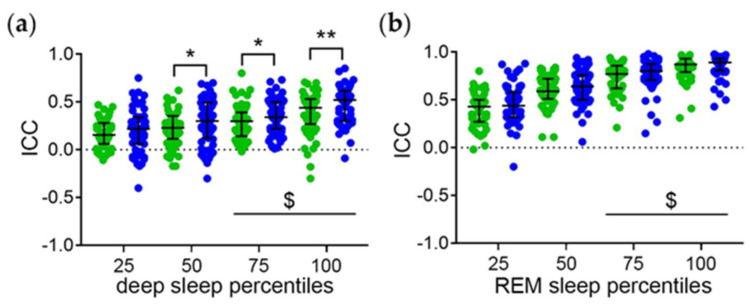
Temporal relationship between the sleep stages and total sleep with regard to the homeostatic component. Intraindividual correlation coefficients (ICC) of midpoint of sleep and the (**a**) deep and (**b**) REM sleep percentiles relative to *minutes after sleep onset*. Workdays (green symbols) and work-free days (blue symbols) were evaluated separately. Middle lines and error bars show median with interquartile range (*n* = 59). Wilcoxon test *, *p* < 0.01; **, *p* < 0.01. $, *p* < 0.05 vs. the respective 25 percentiles.

**Figure 4 brainsci-14-00260-f004:**
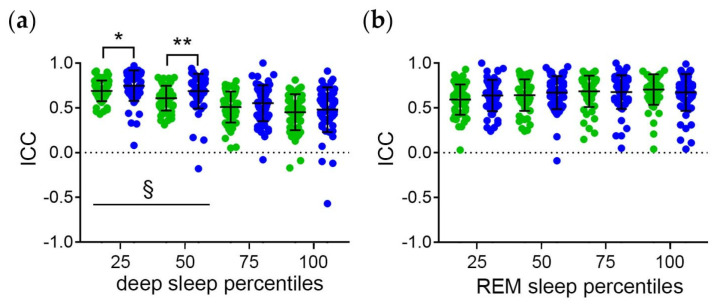
Temporal relationship between the sleep stages and total sleep with regard to the circadian component. Intraindividual correlation coefficients (ICC) of midpoint of sleep and the (**a**) deep and (**b**) REM sleep percentiles relative to *clock time*. Workdays (green symbols) and free days (blue symbols) were evaluated separately to account for workday constraints. Middle lines and error bars show median with interquartile range (*n* = 59). Wilcoxon test *, *p* < 0.01; **, *p* < 0.01. §, *p* < 0.05 vs. the respective 100 percentiles.

**Figure 5 brainsci-14-00260-f005:**
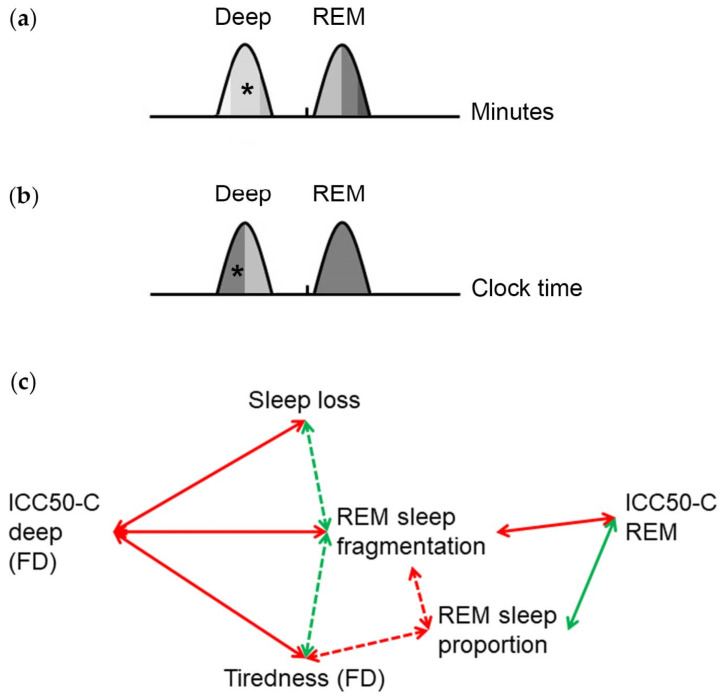
Graphic summary of the temporal relationship between deep and REM sleep with total sleep. Relationship between deep and REM sleep percentiles with the midpoint of total sleep relative to (**a**) the time elapsed since sleep onset, reflecting the homeostatic component and (**b**) clock time, reflecting the circadian component. The intraindividual correlation coefficients (ICC) reflect the strength of the phase relationships. The higher the ICC, the darker the shade under the curve. Asterisks indicate significantly higher ICC on work-free days than on workdays. Phases of the sleep stages are based on our previous study [23]. (**c**) Relationship of the ICC of the 50 sleep percentiles, with regard to the circadian component (ICC50-C), with measures for sleep quality. The ICC50-C *REM* correlate negatively with REM sleep fragmentation and positively with REM sleep proportion. On work-free days (FD), the ICC50-C *deep* correlate negatively with sleep loss, fragmentation of REM sleep and tiredness upon waking. Dashed arrows indicate correlations described in our previous study [23].

**Table 1 brainsci-14-00260-t001:** Relationship between the ICC of the 50 percentiles of REM and deep sleep and the midpoint of sleep in *minutes after sleep onset* (ICC50-H) with measures for sleep quality. Chronotype-dependent sleep loss, tiredness upon waking on work-free days, as well as REM sleep proportion and fragmentation, were used as measures for sleep quality. *n* = 59, *R*, correlation coefficient. Significant correlations are highlighted in italics; *, *p* < 0.05.

	ICC50-H Deep				ICC50-H REM			
	Workdays		Work-Free Days		Workdays		Work-Free Days	
	*R*	*p*	*R*	*p*	*R*	*p*	*R*	*p*
Sleep loss	−0.1	0.45	−0.25	0.056	0.03	0.82	−0.18	0.16
Tiredness	0.15	0.25	−0.15	0.42	−0.04	0.78	0.05	0.73
REM sleep								
Proportion	0.09	0.52	0.1	0.44	*0.29*	*0.025 **	0.21	0.11
Fragmentation	−0.12	0.35	−0.12	0.38	−0.25	0.053	−0.16	0.22

**Table 2 brainsci-14-00260-t002:** Relationship between the ICC of the 50 percentiles of REM and deep sleep and the midpoint of sleep in *clock time* (ICC50-C) with measures for sleep quality. Chronotype-dependent sleep loss, tiredness upon waking on work-free days, as well as REM sleep proportion and fragmentation, were used as measures of sleep quality. *n* = 59, *R*, correlation coefficient. Significant correlations are highlighted in italics; *, *p* < 0.05; **, *p* < 0.01; ***, *p* < 0.001; ****, *p* < 0.0001.

	ICC50-C Deep				ICC50-C REM			
	Workdays		Work-Free Days		Workdays		Work-Free Days	
	*R*	*p*	*R*	*p*	*R*	*p*	*R*	*p*
Sleep loss	0.13	0.34	−*0.49*	*0.00009 *****	0.24	0.07	−0.21	0.11
Tiredness	0.01	0.93	*−0.26*	*0.048 **	0.04	0.78	−0.22	0.09
REM sleep								
Proportion	−0.08	0.53	0.24	0.06	*0.27*	*0.03 **	*0.43*	0.0007 ***
Fragmentation	0.08	0.53	−*0.26*	*0.045 **	−*0.28*	*0.03 **	−*0.39*	*0.002 ***

## Data Availability

The data presented in this study are available on request from the corresponding author. The data are not publicly available due to privacy or ethical restrictions.

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
