# Peer review of "Timing of Deep and REM Sleep Based on Fitbit Sleep Staging in Young Healthy Adults under Real-Life Conditions"

_brainsci, 2024, doi:10.3390/brainsci14030260_

Round 1

Reviewer 1 Report

Comments and Suggestions for Authors

Thank you for the opportunity to review the manuscript, Timing of deep and REM sleep based on Fitbit sleep staging in young healthy adults under real life conditions.  The paper is well-written and its focus, the relative effects of the circadian clock and sleep homeostat on critical sleep parameters, will be of interest to readers of Brain Sciences—and timely, given the attention that sleep quality has received related to mental and physical health.  Most importantly, the paper describes an intriguing experimental design that provides a potential new paradigm for disentangling the effects of the sleep homeostat from circadian clock control of sleep stages and sleep architecture, in general. The paper uses a small, but detailed dataset to provide evidence supporting the strength of the relationship between the homeostatic and circadian components of sleep timing and deep and REM sleep.  In addition, the authors find correlations between the timing of deep and REM sleep and subjective tiredness and REM fragmentation.  Below, I outline a few comments that I hope will help strengthen the impact of the manuscript.

• The use of intraindividual correlation coefficients as a measure to separate circadian and homeostatic effects is very clever, if somewhat difficult to untangle.  The extensive description of the main variables, including the variable ‘relationship/correlation with deep or REM sleep relative to total sleep with regard to the homeostatic/circadian component’ is a bit unwieldy, particularly when this variable is then compared in relationships to other variables.  Could the authors explain this variable description once and then shorten to deep or REM sleep with regard to homeostatic sleep timing (or circadian sleep timing).  Or some such simplification.  This may make it easier for non-specialist readers to absorb the comparisons and understand the interpretations.

• Sleep stage percentiles are a good way to analyze these questions but how these percentiles are calculated should be included in the methods.  They are introduced on line 126 but are not explained until line 157 (in the Results section).  Also, can these descriptions (like in the figure legends for Figure 4 and 5) be shortened to sleep stage percentiles for homeostatic sleep, rather than the long descriptor?

• A similar point to the two above:  Why define a midS and MS and not simply a MS-H (midsleep of homeostat component) and MS-C (midsleep of circadian component)…this would make the interpretation of the results so much easier!  Apologies if this is the standard in the subfield…

• The title of the paper includes ‘…under real life conditions’…but in the methods, we see that the data collection occurred during Covid, which was technically real-life conditions but may have altered sleep patterns, as has been shown in many recent studies.  It would be interesting to see whether these results are repeatable in non-Covid environments.  The authors should address this point more thoroughly given that the real-life conditions may be an outlier (especially since stress can influence the circadian system, line 312).

• In the discussion, the effect of chronotype dependent sleep loss needs more clarity.  

• The results show some significant differences in correlations with both homeostatic and circadian components of sleep…can the authors give a sense of the biological relevance to these differences?  What are the magnitude of differences seen in such sleep architecture variables in unhealthy individuals or individuals with sleep disorders…are some of the chronotype related associations mirroring these magnitudes?  

• Figure 3 shows one subject…would the differences in ICC for deep sleep be this dramatic if the averages were shown? Similarly, it is of interest whether the values for these correlations have a low variance across individuals or have a high variance, with some showing high correlation values and other showing low correlation values (because the interpretation of these for sleep architecture across individuals would be different). For example, in Figure 4, there seem to be more outliers for REM sleep…did the authors consider the patterns of variance in the study?  Perhaps the sample size is too small to have an effective analysis of this…

Minor comments:

• Excellent introduction but it is all one paragraph.  If this is not the required style of the journal, could the authors divide this paragraph into sections for ease in reading…

• Figure 2 is difficult to interpret as the lines for Sat/Sun are not easily visible and there is no legend for the colored dots (representing the percentiles presumably).

• There is a typo in the first line of the legend for Figure 4.

• Figure 6, the graphic summary, is a good idea but it is not as helpful as it could be…could the authors label a and b the homeostat and circadian components…and provide more clarification to section c that these are circadian component effects…

Author Response

Comments and Suggestions for Authors

Thank you for the opportunity to review the manuscript, Timing of deep and REM sleep based on Fitbit sleep staging in young healthy adults under real life conditions.  The paper is well-written and its focus, the relative effects of the circadian clock and sleep homeostat on critical sleep parameters, will be of interest to readers of Brain Sciences—and timely, given the attention that sleep quality has received related to mental and physical health.  Most importantly, the paper describes an intriguing experimental design that provides a potential new paradigm for disentangling the effects of the sleep homeostat from circadian clock control of sleep stages and sleep architecture, in general. The paper uses a small, but detailed dataset to provide evidence supporting the strength of the relationship between the homeostatic and circadian components of sleep timing and deep and REM sleep.  In addition, the authors find correlations between the timing of deep and REM sleep and subjective tiredness and REM fragmentation.  Below, I outline a few comments that I hope will help strengthen the impact of the manuscript.

Reply: Thank you very much for appreciating our experimental paradigm and for the very helpful comments and suggestions.

  • The use of intraindividual correlation coefficients as a measure to separate circadian and homeostatic effects is very clever, if somewhat difficult to untangle.  The extensive description of the main variables, including the variable ‘relationship/correlation with deep or REM sleep relative to total sleep with regard to the homeostatic/circadian component’ is a bit unwieldy, particularly when this variable is then compared in relationships to other variables.  Could the authors explain this variable description once and then shorten to deep or REM sleep with regard to homeostatic sleep timing (or circadian sleep timing).  Or some such simplification.  This may make it easier for non-specialist readers to absorb the comparisons and understand the interpretations.

Reply: We are glad that you like the approach! We now included a small paragraph for each subsection of the results to introduce the major players. This should help the readers orientation. We also moved figure 1 to the supplementary figures so that subsection 3.1. becomes a little slimmer.

  • Sleep stage percentiles are a good way to analyze these questions but how these percentiles are calculated should be included in the methods.  They are introduced on line 126 but are not explained until line 157 (in the Results section).  Also, can these descriptions (like in the figure legends for Figure 4 and 5) be shortened to sleep stage percentiles for homeostatic sleep, rather than the long descriptor?

Reply: We now included the calculation of the sleep stage percentiles in the method section.

  • A similar point to the two above:  Why define a midS and MS and not simply a MS-H (midsleep of homeostat component) and MS-C (midsleep of circadian component)…this would make the interpretation of the results so much easier!  Apologies if this is the standard in the subfield…

Reply: Thank you very much for this suggestion. We now have defined ICC50-H and ICC50-C accordingly.

  • The title of the paper includes ‘…under real life conditions’…but in the methods, we see that the data collection occurred during Covid, which was technically real-life conditions but may have altered sleep patterns, as has been shown in many recent studies.  It would be interesting to see whether these results are repeatable in non-Covid environments.  The authors should address this point more thoroughly given that the real-life conditions may be an outlier (especially since stress can influence the circadian system, line 312).

Reply: This is now discussed in more detail in section 5.

  • In the discussion, the effect of chronotype dependent sleep loss needs more clarity.  

Reply: We revised this paragraph

  • The results show some significant differences in correlations with both homeostatic and circadian components of sleep…can the authors give a sense of the biological relevance to these differences?  What are the magnitude of differences seen in such sleep architecture variables in unhealthy individuals or individuals with sleep disorders…are some of the chronotype related associations mirroring these magnitudes?  

Reply: Our study suggests that stronger phase relationships between total sleep and the sleep stages are associated with better sleep quality in health subjects. To answer this question, further larger cohort studies in healthy subjects and unhealthy individuals are needed. This is a limitation of our study (chapter 5).

Figure 3 shows one subject…would the differences in ICC for deep sleep be this dramatic if the averages were shown? Similarly, it is of interest whether the values for these correlations have a low variance across individuals or have a high variance, with some showing high correlation values and other showing low correlation values (because the interpretation of these for sleep architecture across individuals would be different). For example, in Figure 4, there seem to be more outliers for REM sleep…did the authors consider the patterns of variance in the study?  Perhaps the sample size is too small to have an effective analysis of this…

Reply: Right, there is a difference in the outliers in (former) figures 4 and 5 between REM and deep sleep. Thank you very much for your suggestion, we will keep this in mind for future studies with larger cohorts.  

  • Excellent introduction but it is all one paragraph.  If this is not the required style of the journal, could the authors divide this paragraph into sections for ease in reading…

Reply: Thank you very much. We divided this section into paragraphs.

  • Figure 2 is difficult to interpret as the lines for Sat/Sun are not easily visible and there is no legend for the colored dots (representing the percentiles presumably).

 Reply: This was corrected.

  • There is a typo in the first line of the legend for Figure 4.

  Reply: This was corrected.

  • Figure 6, the graphic summary, is a good idea but it is not as helpful as it could be…could the authors label a and b the homeostat and circadian components…and provide more clarification to section c that these are circadian component effects…

Reply: Thank you very much for this suggestion, which we implemented.

Reviewer 2 Report

Comments and Suggestions for Authors

The manuscript presents a valuable and timely investigation into the temporal relationships between deep sleep, REM sleep, and total sleep as analyzed through Fitbit devices under real-life conditions. The study's focus on the homeostatic and circadian components of sleep, the impact of workday constraints, and the association between sleep stage timing and sleep quality is commendable. The use of wearable technology for sleep tracking in a naturalistic setting adds a novel dimension to sleep research. However, there are several areas where the manuscript could be strengthened to enhance its contribution to the field.

Specific comments are following:

11.  The background could benefit from a more detailed discussion on previous findings related to other DHTs’ role in sleep research.

22. The legends for Figures 4 and 5 describe statistical outcomes (e.g., "Wilcoxon test * P<0.01; ** P<0.01. P<0.05 vs. the respective 25 percentiles.") but could be clearer. Specifically, the use of asterisks and dollar signs to denote significance levels is unconventional. Typically, asterisks are used alone, and their meaning is explained (e.g., "* p < 0.05, ** p < 0.01"). Clarifying this in the figure legend or in a footnote could improve readability and understanding.

33. The term "intra-individual correlation coefficients (ICC)" to describe relationships between sleep stages and total sleep should be consistently used. A brief explanation or reference for the ICC calculation method could enhance comprehension, especially for readers less familiar with this statistical measure.

44.  It’ll help to enhance the quality of discussion by-

·         Providing a more detailed comparison with previous studies.

·         Expanding on the mechanistic insights or hypotheses that might explain the observed sleep patterns.

·         Discussing the implications of the findings for sleep interventions and how they might be applied in real-life settings.

·         Addressing the study's limitations more comprehensively and outlining potential future research directions.

·         Clarifying the novel contributions of the study, particularly in terms of its real-life application and the use of wearable technology.

56.    The conclusions are well-supported by the results. Emphasizing the practical implications of the findings for improving sleep quality in real-life contexts could make the conclusions even more impactful.

Comments on the Quality of English Language

·       1.  Some sentences could be revised for clarity and conciseness. For example, "This suggests in terms of homeostatic timing later deep and REM sleep is stronger associated to total sleep than the earlier deep and REM sleep" might be clearer as "This suggests that, in terms of homeostatic timing, deep and REM sleep later in the night are more strongly associated with total sleep duration than earlier occurrences of these stages.

·      2.    "workday constraints" has been used multiple times but does not provide a detailed explanation of what this entails or how it was quantified in the study.

·        3.  Use of parentheses and commas in descriptions (as seen in the figure legends and explanations) sometimes makes sentences cumbersome.

Author Response

The manuscript presents a valuable and timely investigation into the temporal relationships between deep sleep, REM sleep, and total sleep as analyzed through Fitbit devices under real-life conditions. The study's focus on the homeostatic and circadian components of sleep, the impact of workday constraints, and the association between sleep stage timing and sleep quality is commendable. The use of wearable technology for sleep tracking in a naturalistic setting adds a novel dimension to sleep research. However, there are several areas where the manuscript could be strengthened to enhance its contribution to the field.

Specific comments are following:

  1. The background could benefit from a more detailed discussion on previous findings related to other DHTs’ role in sleep research. Reply: We are afraid that we are not familiar with the abbreviation DHT other than dihydrotestosterone. Could you please specify the studies your are referring to?

  1. The legends for Figures 4 and 5 describe statistical outcomes (e.g., "Wilcoxon test * P<0.01; ** P<0.01. P<0.05 vs. the respective 25 percentiles.") but could be clearer. Specifically, the use of asterisks and dollar signs to denote significance levels is unconventional. Typically, asterisks are used alone, and their meaning is explained (e.g., "* p < 0.05, ** p < 0.01"). Clarifying this in the figure legend or in a footnote could improve readability and understanding. Reply: The meaning of the dollar and paragraph sign is described in the legend.

  1. The term "intra-individual correlation coefficients (ICC)" to describe relationships between sleep stages and total sleep should be consistently used. A brief explanation or reference for the ICC calculation method could enhance comprehension, especially for readers less familiar with this statistical measure. We now use the term consistently and included the calculation of the ICC in the method section
  2. It’ll help to enhance the quality of discussion by-
  • Providing a more detailed comparison with previous studies.
  • Expanding on the mechanistic insights or hypotheses that might explain the observed sleep patterns.
  • Discussing the implications of the findings for sleep interventions and how they might be applied in real-life settings.
  • Addressing the study's limitations more comprehensively and outlining potential future research directions.
  • Clarifying the novel contributions of the study, particularly in terms of its real-life application and the use of wearable technology.

Reply: Thank you very much. We have included an additional paragraph in the discussion

  1.   The conclusions are well-supported by the results. Emphasizing the practical implications of the findings for improving sleep quality in real-life contexts could make the conclusions even more impactful.

Comments on the Quality of English Language Reply: the manuscript will undergo additional language revision.

  •      1.  Some sentences could be revised for clarity and conciseness. For example, "This suggests in terms of homeostatic timing later deep and REM sleep is stronger associated to total sleep than the earlier deep and REM sleep" might be clearer as "This suggests that, in terms of homeostatic timing, deep and REM sleep later in the night are more strongly associated with total sleep duration than earlier occurrences of these stages.
  •     2.    "workday constraints" has been used multiple times but does not provide a detailed explanation of what this entails or how it was quantified in the study. Reply: we explained this now in lanes 144-145.

  •       3.  Use of parentheses and commas in descriptions (as seen in the figure legends and explanations) sometimes makes sentences cumbersome.

Reviewer 3 Report

Comments and Suggestions for Authors

I would recommend the publication of the article after appropriate integrations and a careful revision. The innovative elements and interesting perspectives on the relationship between sleep stages, work constraints, and sleep quality provide a significant contribution to the research field.

However, it is important to address limitations such as the limited sample size and potential issues related to the use of Fitbit algorithms. Integrations with additional data or greater diversity in the sample could enhance the external validity of the study.

Furthermore, a comprehensive review of the updated scientific literature could further strengthen the theoretical context of the work. Industry experts can provide valuable advice on methodology and contribute to enhancing the overall robustness of the article.

After making these integrations and addressing the raised concerns, the article could become a significant and worthy contribution to be published in a scientific journal.

- why did you choose to use a Fit Bit device and not another device?

- do you think the study could be applied to other devices in the future? (e.g. Apple Watch, Garmin etc)

- will the data be usable on a medical level in the future?

Comments on the Quality of English Language

The English used is appropriate

Author Response

I would recommend the publication of the article after appropriate integrations and a careful revision. The innovative elements and interesting perspectives on the relationship between sleep stages, work constraints, and sleep quality provide a significant contribution to the research field.

Reply: Thank you very much!

However, it is important to address limitations such as the limited sample size and potential issues related to the use of Fitbit algorithms. Integrations with additional data or greater diversity in the sample could enhance the external validity of the study.

Reply: We addressed these issues in section 5.

Furthermore, a comprehensive review of the updated scientific literature could further strengthen the theoretical context of the work. Industry experts can provide valuable advice on methodology and contribute to enhancing the overall robustness of the article.

Reply: Thank you very much. We have included an additional paragraph in the discussion. Since the sleep staging by wearables is a new technology, there is limited comparable work in young healthy subjects.

After making these integrations and addressing the raised concerns, the article could become a significant and worthy contribution to be published in a scientific journal.

- why did you choose to use a Fit Bit device and not another device?

- do you think the study could be applied to other devices in the future? (e.g. Apple Watch, Garmin etc)

- will the data be usable on a medical level in the future?

Reply: we now addressed these issues in an additional paragraph in the discussion.